# Dynamic MRI in Degenerative Cervical Myelopathy: A Systematic Review of Radiological Markers, Correlations, and Outcomes

**DOI:** 10.3390/jcm15010265

**Published:** 2025-12-29

**Authors:** Ali Baram, Jad El Choueiri, Carlo Brembilla, Francesca Pellicanò, Lorenzo De Rossi, Leonardo Di Cosmo, Mario De Robertis, Emanuele Stucchi, Donato Creatura, Gabriele Capo, Maurizio Fornari, Marco Riva, Letterio S. Politi, Federico Pessina

**Affiliations:** 1Department of Neurosurgery, IRCCS Humanitas Research Hospital, Via Alessandro Manzoni 56, Rozzano, 20089 Milan, Italy; carlo.brembilla@humanitas.it (C.B.); mario.derobertis@humanitas.it (M.D.R.); gabriele.capo@humanitas.it (G.C.); maurizio.fornari@yahoo.it (M.F.); federico.pessina@hunimed.eu (F.P.); 2Humanitas University, Via Rita Levi Montalcini 4, Pieve Emanuele, 20072 Milan, Italy; jad.elchoueiri@st.hunimed.eu (J.E.C.); francesca.pellicano@st.hunimed.eu (F.P.); lorenzo.derossi@st.hunimed.eu (L.D.R.); leonardo.dicosmo@st.hunimed.eu (L.D.C.); emanuele.stucchi@humanitas.it (E.S.); donato.creatura@humanitas.it (D.C.); 3Department of Biomedical Sciences, Humanitas University, Pieve Emanuele, 20072 Milan, Italy; letterio.politi@hunimed.eu; 4Neuroradiology Department, IRCCS Humanitas Research Hospital, Via Alessandro Manzoni 56, Rozzano, 20089 Milan, Italy

**Keywords:** cervical myelopathy, dynamic MRI, cervical spine

## Abstract

**Background/Objectives**: Conventional static magnetic resonance imaging may underestimate the severity of cervical cord compression by failing to account for positional changes in the spinal canal. Dynamic MRI (dMRI) captures cervical motion, allowing evaluation of cord compression under physiological loading. This systematic review aimed to synthesize evidence on how dMRI modifies the assessment of spinal canal narrowing and signal change, and how these findings correlate with impairment and postoperative outcomes in degenerative cervical myelopathy. **Methods**: A systematic literature search was conducted across PubMed, Scopus, and Embase databases according to PRISMA guidelines. Studies evaluating the role of dMRI (flexion–extension MRI) in diagnosing or predicting outcomes of cervical degenerative pathology were included. Data were extracted on imaging protocols, diagnostic findings, quantitative parameters, and clinical outcomes. **Results**: Nineteen studies met the inclusion criteria. dMRI consistently revealed motion-dependent stenosis and intramedullary signal changes not visible on static imaging. Extension imaging frequently demonstrated disease progression, showing altered spinal cord area, cerebrospinal fluid (CSF) reserve, and additional compression levels. Dynamic sequences enhanced sensitivity for pathological segment detection and improved correlation with clinical severity. Preoperative dMRI findings, particularly extension-related compression and T2 hyperintensity, predicted postoperative neurological recovery and influenced surgical planning in up to one third of cases. **Conclusions**: Dynamic MRI provides superior diagnostic sensitivity and prognostic information compared with static imaging by revealing motion-induced spinal cord compression and microstructural alterations. It should be considered when clinical findings exceed static MRI severity or when the symptomatic level is uncertain. Standardization of protocols and large prospective studies are needed to define evidence-based clinical indications.

## 1. Introduction

Degenerative conditions in the cervical spine, such as cervical spondylosis, come as a natural consequence of aging, and they have been reported to be found in 95% of patients above 65 years of age [1]. In most patients, these changes remain physiological and asymptomatic, but many experience debilitating symptoms with diverse patient presentations, such as axial neck pain, radiculopathy, myelopathy, or a combination of these clinical profiles [2,3].

Degenerative cervical myelopathy (DCM) is a slowly progressive condition resulting from the compression of the spinal cord or surrounding structures due to degenerative changes over time [4,5]. Symptoms vary, and this condition often presents with a variety of neurological findings which can manifest as a loss of manual dexterity, weakness, increased urinary frequency or hesitance, gait dysfunction, and others [6].

These symptoms are, however, not specific to DCM, and the differential diagnosis might include other neurodegenerative pathologies. Magnetic resonance imaging (MRI) is considered the gold standard to visualize degenerative changes related to DCM [7], relying on signal intensity (SI) changes, and other predictors that correlate with severity and surgical outcome [8]. It is important to emphasize, however, that DCM remains a fundamentally clinical diagnosis.

However, due to limitations of conventional MRI scanners, conventional static MRI often fails in revealing the cause or reflecting the real severity of the pathology [9]. The cervical spine is highly mobile [10], thus its anatomical configurations differ between flexion, extension, and the neutral position. This may alter the degree of canal narrowing and cord deformation, introducing the need for an imaging technique able to show these changes.

Dynamic MRI of the cervical spine presents a possible solution to this problem [11,12], offering a view of the spinal cord in flexion and extension.

This systematic review therefore aims to summarize the current evidence on dynamic cervical MRI in degenerative cervical myelopathy, with a specific focus on how flexion-extension imaging alters the evaluation of the cervical canal narrowing and cord compression compared with static MRI, and on the relationship between dynamic MRI findings, neurological impairment, and clinical scales. The study also assesses the impact of these findings on surgical planning and postoperative outcomes.

## 2. Materials and Methods

### 2.1. Literature Review Design

This systematic review was conducted in accordance with PRISMA guidelines (Appendix A) [13].

### 2.2. Search Strategy

A comprehensive online search was conducted across several medical databases including PubMed, Embase, and Scopus until 15 September 2025. The following terms were searched for in titles, abstracts, and keywords, translated through polyglot accelerator to fit each database:

((“Cervical Spine”[MeSH Terms] OR “Cervical Vertebrae”[MeSH Terms] OR “Cervical Spine”[Title/Abstract] OR “Cervical Spine Disease”[Title/Abstract] OR “Cervical Degenerative Disease”[Title/Abstract] OR “Cervical Degeneration”[Title/Abstract] OR “Cervical Spondylosis”[Title/Abstract] OR “Cervical Disc Disease”[Title/Abstract] OR “Cervical Disc Degeneration”[Title/Abstract] OR “Cervical Disc Herniation”[Title/Abstract] OR “Cervical Myelopathy”[Title/Abstract] OR “Cervical Spondylotic Myelopathy”[Title/Abstract] OR “Cervical Stenosis”[Title/Abstract] OR “Cervical Canal”[Title/Abstract] OR “Cervical Cord”[Title/Abstract]))

AND

(“Dynamic MRI”[Title/Abstract] OR “Kinetic MRI”[Title/Abstract] OR “Kinematic MRI”[Title/Abstract] OR “Flexion MRI”[Title/Abstract] OR “Extension MRI”[Title/Abstract] OR “Flexion-Extension MRI”[Title/Abstract] OR “Positional MRI”[Title/Abstract] OR “Upright MRI”[Title/Abstract] OR “Standing MRI”[Title/Abstract] OR “Weight-bearing MRI”[Title/Abstract] OR “Axially Loaded MRI”[Title/Abstract] OR “Loaded MRI”[Title/Abstract] OR “Motion MRI”[Title/Abstract] OR “Dynamic Imaging”[Title/Abstract] OR “Motion-dependent MRI”[Title/Abstract])

AND

(“Spinal Cord Compression”[MeSH Terms] OR “Spinal Stenosis”[MeSH Terms] OR “Spinal Cord”[Title/Abstract] OR “Cord Compression”[Title/Abstract] OR “Cord Pathology”[Title/Abstract] OR “Cord Compromise”[Title/Abstract] OR “Stenotic Changes”[Title/Abstract] OR “Canal Narrowing”[Title/Abstract] OR “Cord Deformation”[Title/Abstract] OR “Cord Signal Change”[Title/Abstract] OR “T2 Hyperintensity”[Title/Abstract] OR “Neurological Deficit”[Title/Abstract] OR “Cord Dysfunction”[Title/Abstract] OR “Neurological Impairment”[Title/Abstract] OR “Myelopathy”[Title/Abstract] OR “Clinical Correlation”[Title/Abstract] OR “Clinical Outcome”[Title/Abstract] OR “Prognosis”[Title/Abstract] OR “Surgical Outcome”[Title/Abstract] OR “Treatment Planning”[Title/Abstract]))

### 2.3. Eligibility Criteria

Our review inclusion criteria were as follows:

(1) Randomized controlled trials, prospective cohort studies, retrospective cohort studies, and case series with more than 10 patients (2) published in English, (3) specifically evaluating dynamic MRI in cervical degenerative diseases with a focus on myelopathy.

Exclusion criteria were as follows: (1) Case reports or series with less than 10 patients involved, reviews, editorials, and letters, (2) studies on non-degenerative spinal conditions (traumas, tumors, infection, or congenital anomalies), (3) cadaveric or animal studies, and (4) overlapping datasets.

### 2.4. Data Extraction and Synthesis

Two researchers (L.D.R. and J.E.C.) independently performed title and abstract screening. This was followed by a full-text screening (J.E.C. and F.P.), and conflicts were resolved by a senior author (A.B.)

Four researchers (J.E.C., L.D.C., F.P., L.D.R.) independently performed the data extraction. A senior independent investigator resolved any inconsistencies. Qualitative information about study population, interventions, and outcomes were then collected and organized in tables.

### 2.5. Quality and Bias Assessment

The methodological quality and risk of bias of the included studies were evaluated using the Methodological Index for Non-Randomized Studies (MINORS) tool [14]. This instrument is specifically designed for both comparative and non-comparative cohort studies, and each item is scored from 0 (not reported) to 2 (adequately reported), with higher scores indicating better methodological quality. More specifically, all 12 items are used for comparative studies, whereas only 8 are used for non-comparative ones.

## 3. Results

Our initial search identified 404 records across three databases. After removing 177 duplicated records, we screened the abstracts of the remaining 227 articles, excluding 168 that did not meet the inclusion criteria. The full-text screening of 59 articles resulted in the exclusion of 40 other studies. Ultimately, 19 studies met the inclusion criteria and were included in the final synthesis. A summary of our search strategy is found in Figure 1. All studies were of moderate or high quality (Table 1).

The individual MINORS scores for the risk of bias assessment of each study are reported in Table 1.

A total of 19 studies, including 10 retrospective and 9 prospective cohorts, met the inclusion criteria, totaling 1638 patients (Table 2). Dynamic MRI was performed using 1.5-T systems and 3-T systems, commonly in the supine position. All protocols included neutral, flexion, and extension sequences, although the range of motion achieved slightly changed based on patient and protocol-specific variations. Among the pathologies investigated, cervical spondylotic myelopathy (CSM) was the predominant diagnosis, while several studies also examined ossification of the posterior longitudinal ligament (OPLL), cervical disc herniation (CDH), subaxial subluxation (SS), and cervical spondylotic radiculopathy (CSR). Comparator imaging was routinely static MRI, serving as the reference for the results derived from dMRI.

### 3.1. Diagnostic Features

#### 3.1.1. Specific Radiological Features

Dynamic MRI revealed position-dependent radiological changes in cervical canal morphology, cord dimensions, and intramedullary signal intensity that were often undetectable on static imaging. Several studies consistently demonstrated that extension was the position most strongly associated with canal narrowing and stenosis progression. Shin et al. [16] reported that extension reduced sagittal cord cross-sectional area (516.95 ± 106.20 mm^2^ vs. 534.04 ± 86.96 mm^2^ in neutral and 560.70 ± 90.53 mm^2^ in flexion) and cerebrospinal fluid (CSF) reserve ratio (*p* = 0.004), while Narvekar et al. [17] found that 35.39% of disc levels increased in stenosis grade during extension (with 25.75% and 27.56% progressing from grade 0/1 to grades 2 and 3, respectively), and that 32.52% of patients showed a mean increase of 1.55 ± 0.75 levels of compression on extension MRI when compared to neutral MRI. Similarly, Harada et al., Makhchoune et al., and Şerifoğlu et al. [4,26,28] all observed additional compression levels revealed in extension, most frequently at C5–C6 and C6–C7. Sun et al. [29] confirmed that extension significantly reduced canal diameters at the narrowest level, producing stenosis in over 40% of cases. Dalbayrak et al. [31] and Zhang et al. [32] further quantified these effects, showing that extension narrowed the canal by 13.4% on average and represented the position with the smallest available space and highest frequency of cord impingement. Kong et al. [25] reinforced the link between dynamic compression and signal change, reporting that extension produced greater compression in patients with intramedullary high-intensity signal (HIS), and compression extension/flexion ratios >1.4 correlated with HIS presence.

With regard to flexion, Žídek et al. [15] showed that patients with mild DCM had markedly reduced elongation of cord length compared with healthy controls, with Muhle grade increasing during flexion in half of the patients. Zeitoun et al. [22] reported that flexion detected hyperintense intramedullary lesions in 10% of cases that were invisible on neutral and extension sequences, while Zhang et al. [32] found that flexion revealed high-intensity signals in 40% of patients. Tykocki et al. [21] added that both a smaller SC area in extension (<55 mm^2^) and a smaller SC+CSF area in flexion (<99 mm^2^) were predictive of intramedullary high signal.

Beyond morphometry and signal, Berberat et al. [24] used diffusion tensor imaging to show that flexion, but especially extension, influenced apparent diffusion coefficient (ADC), axial diffusivity (AD), radial diffusivity (RD), and fractional anisotropy (FA) values, with extension-based ADC achieving 100% sensitivity for detecting pathological segments.

#### 3.1.2. Disease Detection and Reclassification

Dynamic MRI demonstrated higher sensitivity in detecting radiological features associated with disease severity compared with conventional imaging. In a prospective cohort of 191 patients, Shin et al. [16] reported that dMRI revealed cord compression and intramedullary T2 hyperintensity not visible on static MRI, both correlating with lower preoperative functional scores and poorer postoperative recovery. In a cohort of 45 patients undergoing ACDF, Li et al. [18] found significant reductions in canal diameter, increased vertebral translation, and greater posterior disc opening during flexion and extension compared with neutral positioning, suggesting dMRI found pathological changes not available from conventional MRI. These dynamic changes corresponded with neurological impairment, as patients demonstrating motion-dependent stenosis or intramedullary signal change typically presented with lower JOA or Nurick scores, indicating more advanced clinical myelopathy. Pratali et al. [20] (2019) further showed an increased diagnostic consensus by comparing radiological diagnoses made on a series of 18 patients, finding higher interobserver reliability for both morphometric parameters and canal stenosis diagnosis compared to static MRI, with intraclass correlation coefficients exceeding 0.9 for most measures. This improved diagnostic agreement more closely aligned radiological findings with patients’ presenting symptoms, helping distinguish clinically significant compression from incidental degenerative changes.

Zeitoun et al. [22] evaluated 255 cervical levels in patients with spondylotic myelopathy and found that from C3 to C6, around 22.5% of stage 3 levels in extension were originally classified as stage 1 in the neutral position. Furthermore, they reported a higher sensitivity of HIL lesions when performing dMRI in the flexed position, whereby in 10% of patients, HILs were identified only in that position. Similarly, Barlett et al. [23], in a cohort of 60 patients with borderline cord compression, found that extension dMRI significantly improved discrimination between myelopathic and non-myelopathic patients compared with standard MRI. Specifically, they found the ratio of cord to CSF area at the point of maximal compression was the strongest predictor of clinical myelopathy. This supports the clinical relevance of dynamic MRI, as radiological severity during extension correlated more accurately with neurological deficits than neutral imaging alone. Berberat et al. [24] identified additional biomarkers for degenerative cervical myelopathy when combining diffusion tensor imaging with dMRI. The apparent diffusion coefficient (ADC) effectively differentiated pathological from control spinal segments, with increased ADC values during extension correlating with intramedullary hyperintensity and early microstructural cord injury. Makhchoune et al. [26] evaluated 24 patients with CSM and found that extension dMRI revealed additional compression levels in 12 patients. Alkosha et al. [27] found that relying on neutral MRI alone missed 16.7–33.3% of surgical candidates, while dynamic MRI revealed occult compressions, reinforcing its value in improving diagnostic accuracy and operative planning.

#### 3.1.3. Anatomical Metrics

Lee et al. [19] searched for additional anatomical metrics associated with myelopathy diagnosis in dMRI, evaluating C2–7 angle and C7 slope at differing positions. They found that greater extension-induced angulation (eC2–7A) and a smaller C7 slope were strongly associated with dynamic stenosis. Specifically, a e–nC2–7A value exceeding 15.4° achieved a diagnostic accuracy of 88.9% (AUC = 0.934) for identifying motion-dependent compression. Similarly, Tykocki et al. [21] compared morphometric parameters across neutral, flexion, and extension MRI in 55 patients with cervical canal stenosis. They found that smaller spinal cord area in extension (<55 mm^2^) and smaller combined CSF + SC area in flexion (<99 mm^2^) were predictive of high intramedullary signal on T2-weighted imaging, providing quantitative cut-offs to stratify patients at increased risk of cord pathology. Dynamic changes in standardized indices, such as the spinal cord occupation ratio (SCOR), space available for the spinal cord (SAC), or spinal ratio, were not homogeneously reported across the included studies and therefore could not be incorporated into our review.

### 3.2. Postoperative Outcomes

#### Clinical Correlation with Radiographic Features

Neurological impairment in DCM typically affects motor, sensory, and gait functions due to the involvement of corticospinal and posterior column pathways. These deficits are reflected in clinical scales such as the JOA, mJOA, and Nurick grade. After cervical myelopathy surgery, motor strength, hand function, and gait typically show the most reliable and significant improvement, while sensory deficits and sphincter dysfunction recover less consistently and to a lesser extent.

Postoperative outcomes across the included studies demonstrated significant neurological improvement following surgical decompression for DCM, though prognosis varied depending on dMRI findings and intramedullary signal characteristics. Shin et al. [16] reported marked functional recovery in 191 patients, with mean JOA scores improving from 11.07 ± 2.41 preoperatively to 14.89 ± 1.75 at 2-year follow-up (recovery rate 85.49 ± 13.21%). Notably, patients with intense preoperative intramedullary high signal intensity on T2WI exhibited significantly lower baseline JOA scores and recovery rates compared to those without SI changes. Correspondingly, Nurick grades improved from a mean of 2.56 ± 0.98 to 1.63 ± 0.87. Şerifoğlu et al. [4] found superior outcomes when dynamic MRI was incorporated into surgical planning, with patients in the dynamic group achieving greater postoperative gains in mJOA scores (11.9 ± 2.4 to 15.8 ± 1.6) and larger reductions in VAS pain scores (7.3 ± 1.6 to 3.2 ± 1.4), both significantly better than the static MRI group (*p* = 0.01).

On the other hand, Sun et al. [29] showed that patients with cord compression detectable only on kinetic MRI achieved comparable improvements in JOA scores and recovery rates to conventional myelopathy patients after ACDF. Seki et al. [30] further demonstrated that postoperative recovery was strongly influenced by flexion MRI findings: patients without high intramedullary signal in flexion achieved significantly higher recovery rates (80.9–75.1%) compared to those with flexion-related hyperintensity (33.2–43.5%, *p* < 0.000005), whereas extension-related hyperintensity alone did not significantly affect outcomes.

## 4. Discussion

Nineteen studies were included in this systematic review, analyzing the diagnostic and prognostic utility of dynamic MRI in cervical degenerative spine pathologies. The findings show that dynamic MRI can reveal pathological changes in cervical stenosis that are often undetectable on static imaging. Extension was most strongly associated with stenosis progression, with studies demonstrating reduced cord cross-sectional area, decreased CSF reserve, and additional levels of compression [16,17,28,29,31,32]. Advanced imaging also confirmed that dynamic changes, particularly during extension, significantly influenced microstructural parameters such as ADC, thereby enhancing sensitivity for detecting pathological segments [24]. Furthermore, dynamic MRI demonstrated prognostic relevance for surgical outcomes: across studies, JOA and mJOA scores improved markedly after surgery, with possible greater functional and pain improvements reported when dynamic MRI was incorporated into surgical planning compared to static imaging alone [4].

Dynamic MRI has been shown to improve both the diagnostic sensitivity and characterization of DCM compared with conventional static imaging. It consistently reveals position-dependent cord compression and intramedullary signal changes that correlate with disease severity and functional decline [16,18]. By exposing occult or motion-induced stenosis, dMRI enables the reclassification of patients who would otherwise appear less severe on neutral imaging [22,23,27]. Furthermore, dMRI has provided the opportunity to introduce several additional morphometric parameters with diagnostic utility. For example, C2–7 angulation and C7 slope at differing positions have been proposed as markers of dynamic instability. In addition, thresholds for spinal cord and CSF area at differing positions have been suggested as tools for risk stratification [21].

The superiority of extension imaging can be explained by cervical spine biomechanics, with the mid-cervical levels (C5-C7) being particularly vulnerable. Flexion, on the other hand, stretches the cord and marginally enlarges the canal, potentially increasing longitudinal strain, and this could explain the flexion-related intramedullary hyperintensity reported in many studies [32]. A salient clinical implication of dMRI was disease reclassification and allowing differention between “radiologically myelopathic” and “radiologically non-myelopathic” patients [24]. These data, incorporated into surgical planning, improve decompression targeting and help prevent under-treatment of motion-dependent lesions. Sosa et al. [33] performed a retrospective study of 90 patients to describe the utility of dynamic MRI in inflammatory and undifferentiated myelopathies with T2-hyperintensity and pointed out that myelopathy with T2-hyperintensity was frequently misdiagnosed as myelitis, often leading to unnecessary immunotherapy and to a delay of necessary treatment. Misdiagnosis happens due to the diverse presentation of DCM, further establishing the role of dynamic MRI in narrowing the differential diagnosis.

Thus, flexion-extension imaging does not only contribute to severity grading within DCM, but also to accurate diagnostic differentiation across the spectrum of cervical myelopathies [33]. Dynamic MRI has shown importance in other pathologies besides DCM. In Hirayama disease, for example, early diagnosis is important to advise patients against neck flexion movements. Flexion MRI thus becomes important, as neutral MRI may miss findings [34]. Similarly, cerebellar herniation in Chiari patients can be dynamic [35] and can be evaluated by dMRI.

As previously mentioned, dMRI not only refines diagnosis but also enhances risk stratification and prognostication. Integrating dMRI into clinical algorithms allows for a more individualized therapeutic approach, since, for example, patients with motion-induced HIS or pronounced extension compression may benefit from earlier surgical decompression, while others with preserved microstructural integrity might be candidates for other therapeutic approaches. dMRI can also affect the surgical approach and outcomes, as evidenced by the improvement of neurological clinical outcomes when dMRI was incorporated in surgical planning.

This imaging technique remains, however, limited by the low-field magnet resulting in a low signal-to-noise ratio with reduced image quality. Similarly, it takes longer, and puts patients in positions that can induce symptoms, and they might find it difficult to maintain the necessary immobility [36]. Many studies have noted that there was not a specific flexion/extension angle used, and that it was different for every patient, so the issue of standardization also arises. These differences underscore the potential need for consensus protocols defining optimal positioning [36].

Future research should prioritize the standardization and quantitative validation of dMRI protocols. Artificial intelligence and automated segmentation hold promise for the objective quantification of spinal cord dynamics and deformation, and this should be further explored. Large-scale, multicentric prospective studies correlating imaging parameters with surgical outcomes and neurological recovery are needed to define thresholds and confirm the prognostic and clinical utility of dMRI.

### Limitations

Our review has several limitations that should be acknowledged. First, most included studies were non-randomized, single-center, and heterogeneous in design, with variability in scanner strength, positioning technique, and range of motion, which limited pooled statistical analysis and precluded a meta-analysis. Second, publication bias cannot be excluded. Finally, the inclusion of English-language articles only may have excluded some relevant data.

## 5. Conclusions

Dynamic MRI represents an evolution in the imaging of cervical degenerative spine pathologies, bridging anatomy and function. By visualizing the spinal canal and cord under physiological motion, it uncovers transient but clinically significant compression that static MRI cannot reveal. dMRI could be used in cases where static imaging does not fully reflect clinical findings, or when severity and distribution of stenosis are uncertain. In surgical planning, where extension-dependent decompression may influence the choice or extent of decompression, it could prove useful, as it could also in the postoperative setting to evaluate residual or recurrent stenosis.

Beyond improving diagnostic sensitivity, dMRI can enhance disease stratification, support differential diagnosis, and predict surgical outcomes.

Standardization of protocols, validation of quantitative biomarkers, and prospective outcome correlation are imperative. As imaging technology advances and artificial intelligence becomes integrated into analysis, dynamic MRI could redefine spinal diagnostics.

## Figures and Tables

**Figure 1 jcm-15-00265-f001:**
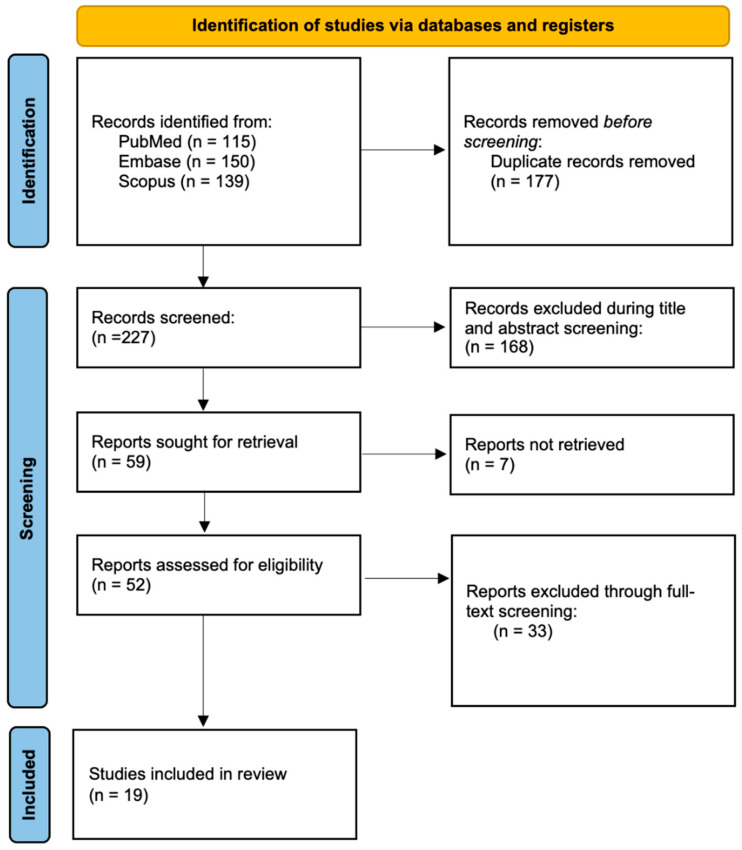
PRISMA flowchart describing the article screening, selection, and extraction process.

**Table 1 jcm-15-00265-t001:** Risk of bias evaluation of the included studies performed with the MINORS scale.

Study	Total MINORS Score
Zidek, 2022	18/24
Shin, 2024	14/16
Narvekar, 2024	12/16
Li, 2023	14/16
Lee, 2021	12/16
Pratali, 2019	14/16
Tykocki, 2018	11/16
Zeitoun, 2015	12/16
Bartlett, 2013	12/16
Berberat, 2023	22/24
Kong, 2025	21/24
Makhchoune, 2022	13/16
Alkosha, 2022	14/16
Harada, 2010	12/16
Şerifoğlu, 2025	18/24
Sun, 2017	19/24
Seki, 2015	12/16
Dalbayrak, 2015	12/16
Zhang, 2011	14/16

**Table 2 jcm-15-00265-t002:** Summary of demographics and MRI parameters of the included studies.

Author, Year	Study Design	Sample Size	Mean Age (±SD)	Pathology	Field (Tesla)	Dynamic MRI Positions			Patient Posture	Comparator
						N	F	E		
Zidek, 2022 [15]	Prospective	Cases: 10 Controls: 10	Cases: 36 Control: 52	Cervical myelopathy	1.5	Y	Y	Y	Prone and supine	Healthy vs. myelopathic
Shin, 2024 [16]	Prospective	191	55.34 ± 12.09	Cervical myelopathy	3	Y	Y	Y	Supine	Static
Narvekar, 2024 [17]	Retrospective	369	50.1 ± 3.8	Cervical myelopathy	1.5	Y	Y	Y	Supine	Static
Li, 2023 [18]	Retrospective	45	56.3	Cervical myelopathy	1.5	Y	Y	Y	Supine	Static
Lee, 2021 [19]	Retrospective	63	68 ± 11.4	Cervical myelopathy	1.5/3	Y	Y	Y	Supine	Static
Pratali, 2019 [20]	Prospective	18	60	Cervical myelopathy	1.5	Y	Y	Y	Supine	Static
Tykocki, 2018 [21]	Retrospective	55	57 ± 13	Cervical myelopathy	1.5	Y	Y	Y	Supine	Static
Zeitoun, 2015 [22]	Retrospective	51	60.3	Cervical myelopathy	1.5	Y	Y	Y	Supine	Static
Bartlett, 2013 [23]	Retrospective	60	55	Cervical myelopathy	1.5	Y	Y	Y	Supine	Static
Berberat, 2023 [24]	Prospective	IHIS+: 10 IHIS−: 11	IHIS+: 55 ± 11 IHIS−: 59 ± 14	Cervical myelopathy	3	Y	Y	Y	NA	Static
Kong, 2025 [25]	Prospective	70	54.97 ± 10.01	Cervical myelopathy	3	Y	Y	Y	NA	Static
Makhchoune, 2022 [26]	Prospective	24	57.9	Cervical myelopathy	1.5	Y	Y	Y	NA	Static
Alkosha, 2022 [27]	Prospective	24	50.1 ± 7.6	Cervical myelopathy	1.5	Y	Y	Y	Supine	Static
Harada, 2010 [28]	Retrospective	54	65.4	Cervical myelopathy	NA	Y	Y	Y	Supine	Static
Şerifoğlu, 2025 [4]	Retrospective	Static: 43 Combined: 39	Static: 55.2 Combined: 56.8	Cervical myelopathy	1.5	Y	Y	Y	Supine	Static
Sun, 2017 [29]	Retrospective	Cases: 31 Control: 31	61.8 ± 9.2	Cervical myelopathy	1.5	Y	Y	Y	Supine	Static
Seki, 2015 [30]	Retrospective	121	65	CSM, OPLL, CDH, SS	1.5	Y	Y	Y	Supine	Static
Dalbayrak, 2015 [31]	Prospective	258	51	CSM, CSR	1.5	Y	Y	Y	Supine	Static
Zhang, 2011 [32]	Prospective	50	60.3	Cervical myelopathy	1.5	Y	Y	Y	Supine	Static

Abbreviations: N: Neutral. F: Flexion. E: Extension. Y: Yes. IHIS: Intramedullary High-Intensity Signal. CSM: Cervical Spondylotic Myelopathy. OPLL: Ossification of the Posterior Longitudinal Ligament. CDH: Cervical Disc Herniation. SS: Subaxial Subluxation. CSR: Cervical Spondylotic Radiculopathy. NA: Not Available.

## Data Availability

The original data presented in the study are openly available in PubMed, Embase, Scopus.

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
