# Peer review of "Dynamic MRI in Degenerative Cervical Myelopathy: A Systematic Review of Radiological Markers, Correlations, and Outcomes"

_jcm, 2025, doi:10.3390/jcm15010265_

Round 1
Reviewer 1 Report
Comments and Suggestions for Authors
Dynamic cervical MRI (dMRI) provides additional diagnostic value over standard static MRI by revealing motion-dependent stenosis, reduced CSF space, and intramedullary T2 changes that are often invisible in neutral position. Across studies, up to one-third of patients show additional compressive levels in extension, and dMRI can refine surgical planning and better correlate imaging with clinical severity, particularly in cases with clinical–radiological mismatch.
However, significant limitations remain. dMRI lacks standardized acquisition protocols, shows substantial methodological heterogeneity, and often suffers from reduced image quality due to low-field systems and patient motion. These factors limit reproducibility and widespread clinical adoption. Importantly, no robust evidence demonstrates that dMRI improves long-term outcomes or cost-effectiveness. The technique requires longer acquisition times, greater patient cooperation, and specialized equipment, increasing procedural costs without clear proof of improved clinical benefit. Consequently, while dMRI is useful in selected and complex cases, its routine use is not currently justified from a cost–benefit perspective.
Author Response
Response to Reviewer 1
To the editor and to reviewer 1,
We have read and appreciated the comments given by reviewer 1. We would first like to thank them for their suggestions, as we believe the changes we made make our manuscript a better fit for the current literature. We appreciate the time and effort put into these revisions, and hereby present a point-by-point clarifications on what we changed
Comment :
Dynamic cervical MRI (dMRI) provides additional diagnostic value over standard static MRI by revealing motion-dependent stenosis, reduced CSF space, and intramedullary T2 changes that are often invisible in neutral position. Across studies, up to one-third of patients show additional compressive levels in extension, and dMRI can refine surgical planning and better correlate imaging with clinical severity, particularly in cases with clinical–radiological mismatch.
However, significant limitations remain. dMRI lacks standardized acquisition protocols, shows substantial methodological heterogeneity, and often suffers from reduced image quality due to low-field systems and patient motion. These factors limit reproducibility and widespread clinical adoption. Importantly, no robust evidence demonstrates that dMRI improves long-term outcomes or cost-effectiveness. The technique requires longer acquisition times, greater patient cooperation, and specialized equipment, increasing procedural costs without clear proof of improved clinical benefit. Consequently, while dMRI is useful in selected and complex cases, its routine use is not currently justified from a cost–benefit perspective.
Response:
We thank the Reviewer for the thoughtful and detailed comment. We agree that dynamic cervical MRI can provide additional diagnostic information compared with standard static MRI, particularly by revealing motion-dependent stenosis, reduction of CSF space, and intramedullary signal changes that may not be evident in the neutral position, especially in cases of clinical–radiological mismatch.
At the same time, we fully acknowledge the important limitations highlighted by the Reviewer. The lack of standardized acquisition protocols, the methodological heterogeneity across studies, and the variability in image quality related to scanner characteristics and patient motion are well-recognized issues. These factors limit reproducibility and currently hinder widespread clinical adoption. For this reason, they have been explicitly discussed in our manuscript as major limitations affecting both routine clinical use and the strength of the available scientific evidence.
We also agree that, at present, robust evidence demonstrating a clear benefit of dMRI in terms of long-term outcomes or cost-effectiveness is still lacking. Longer acquisition times, greater patient cooperation, and higher technical requirements further support the view that dMRI cannot yet be recommended as a routine examination.
These same limitations, however, were the main motivation for performing this systematic review. Given the relevance of the topic and the opportunity provided by the Special Issue, we aimed to critically summarize the current evidence and clarify where dMRI appears to add value and where important gaps remain. In our opinion, the available data—while clearly highlighting the above shortcomings—are nonetheless encouraging for selected and complex cases. When applied with appropriate indications and, ideally, within standardized protocols, dMRI may help refine diagnosis and improve risk stratification. Its integration into clinical decision-making may support a more individualized therapeutic approach.

Reviewer 2 Report
Comments and Suggestions for Authors
Thank you very much for the opportunity to review this interesting article.
This topic is very relevant, as in many clinical cases, spinal canal stenosis is more pronounced during flexion and extension of the cervical spine. This study should take into account the clinical manifestations of cervical myelopathy depending on the degree of narrowing and compression of the spinal canal during cervical flexion and extension in dMRI
The title needs to be changed. It should be more specific and in line with the stated goal and the results obtained.
In introduction:
I really liked the beginning of the introduction with the indication that the degenerative conditions in the cervical spine, such as cervical spondylosis, come as a natural consequence of aging, and it has been reported to be found in 95% of patients above 65 years of age. However, since this is how it started, it may be possible to establish the line between physiological and pathological mechanisms of spondylosis development.
It is necessary to clarify that the diagnosis of Degenerative Cervical Myelopathy (DCM) is fundamentally a clinical diagnosis, relying on a thorough neurological examination that identifies signs of sensory and pyramidal tract dysfunction consistent with spinal cord damage at the level of compression. While magnetic resonance imaging (MRI) is the gold standard for visualizing the morphological signs of spinal cord compression (e.g., spinal canal narrowing, disc herniation, ligament changes), imaging alone confirms the anatomical predisposition, not the clinical manifestation or functional neurological injury. The presence of radiological compression in asymptomatic individuals is common (non-myelopathic spinal cord compression); therefore, a correlation between clinical symptoms and imaging findings is essential for a definitive DCM diagnosis
The purpose needs to be changed first of all, it is unclear secondly, it is not specific.
Materials and methods are described in detail and clearly according with PRISMA guidelines.
I think it would be better and more informative if Table 1 was moved to the results.
In results:
While section 3.1.2 discusses reports on the correlation between the clinical manifestations of Degenerative Cervical Myelopathy (DCM) and morphological changes in the spinal cord, it is necessary to present these findings in a clear, comparative format to help readers distinguish between radiological changes and actual clinical impact. Please provide an additional table for these clinical changes or correlations.
Please add more information about the nature of neurological impairment in DCM with the degree of damage to sensory or motor functions in patients with spinal stenosis and what is the degree of recovery of these impairments after surgical decompression of the spinal canal.
Could you please tell me if the review articles you selected contain information on changes in Torg-Pavlov ratio, Spinal Cord Occupation Ratio (SCOR), Space Available for the Cord and spinal cord flattening ratio according to MRI data in response to extension or flexion of the cervical spine?
Add reference to line 296, 298, 306, 309, 311, 319, 321, 323, 328
The discussion and conclusions should be modified according to changes in the results.
References are not formatted by journal rules
Overall, the topic is very interesting and, as I've already noted, highly relevant. However, the authors focused only on general information about spinal canal narrowing during cervical flexion and extension and the progression of myelopathy. If the authors had been able to identify more criteria for spinal cord compression and establish a link between spinal cord compression and the degree of damage to sensory and motor fibers, it would have been possible to identify risk factors for myelopathy and the prognosis of the disease.
Author Response
Response to Reviewer 2
To the editor and to reviewer 2,
We have read and appreciated the comments given by reviewer 2. We would first like to thank them for their suggestions, as we believe the changes we made make our manuscript a better fit for the current literature. We appreciate the time and effort put into these revisions, and hereby present a point-by-point clarifications on what we changed
Comment 1: Title
Response: We thank the reviewer for this valuable suggestion, and we fully agree. To better reflect the scope and findings of our work, we have changed the title to: “Dynamic Cervical MRI in Degenerative Cervical Myelopathy: Systematic Review of Radiological Markers, Correlations, and Outcomes” (page 1).
Comment 2: Purpose
Response: We agree and have clarified the objective of the review both in the Abstract and in the Introduction. The Abstract now states: “This systematic review aimed to synthesize evidence on how dMRI modifies the assessment of spinal canal narrowing and signal change, and how these findings correlate with impairment and postoperative outcomes in degenerative cervical myelopathy.” (Abstract, page 1). In the final paragraph of the Introduction we now specify three predefined aims related to (1) radiological assessment, (2) clinical correlation, and (3) impact on surgical planning and outcomes (page 2, lines 74-79).
Comment 3: Introduction clarification (DCM as a clinical diagnosis, physiologic vs pathological degeneration)
Response: This is an important point and we have revised the Introduction accordingly. We now explicitly state that DCM is fundamentally a clinical diagnosis based on neurological examination (page 2 lines 62-63), and that MRI demonstrates anatomical changes rather than functional impairment alone. In addition, we expanded the opening paragraph to distinguish age-related physiological cervical spondylosis from pathological degenerative changes that lead to DCM (page 2, lines 48-49).
Comment 4: Move Table 1 to Results
Response: As suggested, we have moved Table 1 (MINORS risk-of-bias assessment) from the Methods section to the Results section, immediately after the description of study selection and quality (pages 5-6, Table 1).
Comment 5: Clinical–radiological correlations in a comparative format
Response: We thank the reviewer again for the helpful suggestion. We considered adding a dedicated table summarizing clinical-radiological correlations, however the included studies used highly heterogeneous clinical scales and reported correlations in varying formats, which would limit comparability and risk oversimplification. Instead, we enhanced clarity as requested, by revising the Results sections 3.1.2 and 3.2.1 (pages 9-10) to describe, in a better structured manner, how dynamic MRI findings can correlate with neurological impairment and postoperative recovery. We believe this approach improves readability and preserves accuract.
Comment 6: More information about neurological impairment and recovery
Response: We have expanded the Results to better describe the nature of neurological impairment in DCM and the degree of recovery after decompression. In section 3.2.1, we added text summarizing the different symptoms of DCM, and recovery after surgery (page 10, lines 267-272)
Comment 7: Torg–Pavlov ratio, SCOR, SAC, flattening ratio
Response: We thank the reviewer for this helpful observation. Some of the included studies did report individual quantitative indices such as the Torg–Pavlov ratio, SCOR, SAC, or related morphometric parameters; however, these measures were not reported homogeneously across studies, with each study using different definitions, measurement techniques, or reporting formats. This heterogeneity precluded their inclusion in a pooled or comparative analysis.
To clarify this for readers, we added a statement in Section 3.1.3 (Anatomical Metrics) indicating that although some of these variables were described, their inconsistent reporting prevented structured integration into our results. (page 10, lines 259-263)
Comment 8: Add references at lines 296, 298, 306, 309, 311, 319, 321, 323, 328
Response: We thank the reviewer for pointing this out. We have added or updated references supporting each of the statements in these lines, including Sosa et al. for dynamic MRI in inflammatory/undifferentiated myelopathies, Paladi et al. for Hirayama disease, Tietze et al. for dynamic Chiari herniation, and Michelini et al. for the state-of-the-art review on dynamic spine MRI.
Comment 9: Discussion and conclusions should be modified according to changes in results
Response: Our discussion conclusions now better reflect our results, that are clearer themselves. Thank you for that.
Comment 10: References not formatted by journal rules
Response: We have carefully rechecked the reference list, it complies with the Journal of Clinical Medicine reference style, standardizing author order, article titles, journal abbreviations, year/volume/page format, and inclusion of DOIs where available.
We thank the reviewer once again for their well-thought comment. We believe the updated version fits the literature better, and we are confident we managed to improve the manuscript thanks to their comments.
Thank you.

Round 2
Reviewer 2 Report
Comments and Suggestions for Authors
Thank you for the opportunity to review this manuscript again.
The change of title and purpose, in my opinion, is very successful and clarifies the essence of the work.
The subsequent revisions and supplemental data integrated into the introduction have significantly enhanced its scholarly value and improved reader engagement.
Changes in materials and methods added more clarity to the work carried out.
Changes in the discussion and conclusions made these sections more specific and related to the results obtained.
In my opinion, the revised version is significantly improved.